# Yield Increase and Emission Reduction Effects of Alfalfa in the Yellow River Irrigation District of Gansu Province: The Coupling Mechanism of Biodegradable Mulch and Controlled-Release Nitrogen Fertilizer

**DOI:** 10.3390/plants14132022

**Published:** 2025-07-02

**Authors:** Wenjing Chang, Haiyan Li, Yaya Duan, Yi Ling, Jiandong Lu, Minhua Yin, Yanlin Ma, Yanxia Kang, Yayu Wang, Guangping Qi, Jianjun Wang

**Affiliations:** 1College of Water Conservancy and Hydrpower Engineering, Gansu Agricultural University, Lanzhou 730070, China; 1073324120789@st.gsau.edu.cn (W.C.); 107332201100@st.gsau.edu.cn (H.L.); 1073323020376@st.gsau.edu.cn (Y.D.); 1073324120786@st.gsau.edu.cn (Y.L.); 1073324120819@st.gsau.edu.cn (J.L.); yanxiakang@gsau.edu.cn (Y.K.); wangyy@gsau.edu.cn (Y.W.); qigp@gsau.edu.cn (G.Q.); 20212201021@st.gsau.edu.cn (J.W.); 2Gansu Province Jingtai Chuan Power Irrigation Water Resource Utilization Center, Baiyin 730400, China

**Keywords:** controlled-release nitrogen fertilizer, biodegradable mulch, N_2_O emission, planting pattern, nitrogen management, yield, alfalfa

## Abstract

Agricultural production in Northwest China is widely constrained by residual plastic film pollution, excessive greenhouse gas emissions, and low productivity. Integrating biodegradable film with controlled-release nitrogen fertilizer offers a promising approach to optimize crop management, enhance yield, and improve environmental outcomes. In this study, three planting patterns (conventional flat planting, FP; ridge mulching with biodegradable film, BM; and ridge mulching with conventional plastic film, PM), two nitrogen fertilizer types (urea, U, and controlled-release nitrogen fertilizer, C), and four nitrogen application rates (0, 80, 160, and 240 kg·hm^−2^) were applied to systematically investigate their effects on alfalfa yield and N_2_O emissions from grasslands. The results showed that BM and PM increased alfalfa yield by 23.49% and 18.65%, respectively, compared to FP, while C increased yield by 8.46% compared to urea. The highest yield (24.84 t·hm^−2^) was recorded under the BMC2 treatment, which was 97.11% higher than that of FPN0. N_2_O emission flux and cumulative emissions increased with nitrogen application rate. Compared with U, C reduced cumulative N_2_O emissions and greenhouse gas emission intensity (GHGI) by 23.89% and 25.84%, respectively. Compared to PM, BM reduced cumulative N_2_O emissions and GHGI by 11.58% and 20.15%, respectively. Principal component analysis indicated that the combination of ridge mulching with biodegradable film and 160 kg·hm^−2^ of C was optimal for simultaneously increasing alfalfa yield and reducing N_2_O emissions, making it a suitable planting–fertilization strategy for the Yellow River irrigation district in Gansu and similar ecological regions.

## 1. Introduction

Global climate warming continues to intensify, posing serious challenges to human society and ecosystems [1]. Greenhouse gas (GHG) emissions are the primary driver of global warming. Among the three major GHGs-CO_2_, CH_4_, and N_2_O, N_2_O has a global warming potential (GWP) 298 times higher than CO_2_ over a 100-year time scale [2]. According to the IPCC AR6 (2023) report, anthropogenic N_2_O emissions account for approximately 40% of total global emissions, with agriculture contributing more than 70% [3]. In agricultural production systems, N_2_O emissions are closely linked to soil nitrification and denitrification processes regulated by biological and environmental factors, with environmental effects largely determined by field management practices [4]. However, current agricultural production is often constrained by extensive field management, simplistic planting patterns, and a focus on productivity at the expense of ecological outcomes [5]. Therefore, optimizing cropping management to synergistically enhance both productivity and environmental sustainability is essential for promoting agricultural transformation and high-quality development.

Nitrogen is a vital nutrient for crop growth and development. Rational nitrogen fertilization is key to replenishing soil nitrogen reserves and ensuring high crop productivity [6]. In contrast, improper N application can lead to nitrate accumulation in plant tissues; disrupt nitrogen metabolism; reduce photosynthetic efficiency [7]; and result in low fertilizer use efficiency, resource waste, soil acidification, compaction, and increased N_2_O emissions [8]. To improve nitrogen use efficiency (NUE), extensive research has focused on developing novel nitrogen fertilizers, improving application techniques, and optimizing planting systems. Studies have shown that, compared with conventional urea, new-generation nitrogen fertilizers, such as slow-release fertilizers, biofertilizers, and stabilized nitrogen fertilizers (containing nitrification and urease inhibitors), can achieve stable and sustained nutrient release via physical coating, chemical inhibition, or biological activation, thereby enhancing root growth and crop yield. These fertilizers also reduce substrate concentrations for nitrification and lower agricultural N_2_O emissions [9]. In addition, liquid nitrogen fertilizers and organic–inorganic compound fertilizers have demonstrated good performance in agricultural production [10,11]. Significant advances have also been made in nitrogen application technologies, including split application, side-dressing, deep placement, and precision fertilization. Compared with broadcast application, deep placement or hole fertilization can enhance ammonium (NH_4_^+^) adsorption by soil particles, reduce NH_4_^+^-N diffusion, and inhibit nitrification at localized high concentrations, thereby prolonging fertilizer effectiveness and reducing N losses [12,13]. Split application aligns nitrogen supply with crop demand at key growth stages, significantly improving NUE [14]. Spectral-based variable rate fertilization, using multispectral and hyperspectral sensors combined with machine learning algorithms, can monitor crop canopy reflectance in real time to diagnose nitrogen status and generate prescription maps, enabling precision fertilization at the field scale and reducing nitrogen leaching and volatilization [15]. Water-fertilizer integration technology, as exemplified by Israel, achieves precise supply of nutrients through pressure-compensated drip heads, resulting in a nitrogen fertilizer utilization rate of 60~80% and a significant reduction in agricultural surface pollution [16]. Planting patterns also regulate the spatial distribution of light, heat, water, and nutrients, influencing microbial abundance and activity and thus driving soil nitrogen cycling processes [17]. Studies have shown that, compared with flat planting, ridge planting improves rhizosphere conditions, promotes microbial proliferation and nutrient transformation, and enhances crop yield as well as water and light use efficiency [18]. When combined with field mulching, ridge planting improves surface energy balance and water redistribution. Designing the structure of a ridge–furrow system is a key aspect of ridge cultivation technology. Zhang et al. [19] found that when the ridge–furrow ratio was 60 cm:60 cm, the rainwater collection efficiency was significantly higher than that of traditional earth ridges on the Loess Plateau, China. Liu et al. [20] found that when the ridge–furrow ratio was 75 cm:50 cm, it could increase the photosynthetic rate and yield of wheat on the Huang Huaihai Plain, China. Ridge cultivation combined with plastic mulching effectively reduces soil moisture evaporation, promotes the convergence of rainfall on the ridge surface into the furrows, increases the infiltration depth and water-holding capacity of soil moisture in the furrows, and forms a synergistic mechanism of dynamic water exchange and complementarity between ridges and furrows. In the arid Loess Plateau, the fully mulched ridge–furrow system increases precipitation use efficiency to over 70%, 20~25% higher than flat planting [21]. In the black soil region of Northeast China, ridge mulching increases effective accumulated temperature and advances maize phenology by 7~10 days [22]. Straw mulching, by providing physical insulation, buffers environmental fluctuations, stabilizes soil temperature and moisture, and suppresses N_2_O emission spikes triggered by drying–rewetting cycles [23]. In recent years, environmentally friendly mulching materials such as liquid films and biodegradable films have also been increasingly applied in agricultural production [24].

In summary, while substantial progress has been made in optimizing nitrogen management and cropping systems, most studies have focused on either fertilization or planting pattern as a single factor. Few have addressed the synergistic effects of combined planting–fertilization regimes on crop performance and N_2_O emissions, and most existing studies focus on annual food and cash crops [25,26,27], with limited research on perennial forage crops, especially alfalfa. Alfalfa (*Medicago sativa* L.), the most widely cultivated perennial legume forage worldwide, is known as the “Queen of Forages” due to its high protein content, strong adaptability, and excellent ecological benefits [28]. However, its efficient nitrogen fixation exacerbates nitrogen management complexity, and N_2_O emission fluxes from alfalfa systems are generally higher than those of other crops [29]. With abundant sunshine and large diurnal temperature variation, Gansu Province offers ideal conditions for alfalfa cultivation and has become one of China’s primary high-quality alfalfa production regions [30]. Given this context, this study systematically investigated the effects of planting pattern, nitrogen fertilizer type, and nitrogen application rate on alfalfa yield and N_2_O emissions, with the objectives to (1) quantify the differences in N_2_O emission indicators under different planting–fertilization regimes; (2) identify the key factors influencing N_2_O emissions from alfalfa grasslands; and (3) establish a planting–fertilization model for achieving yield enhancement and emission reduction in alfalfa systems.

## 2. Results and Analysis

### 2.1. Alfalfa Yield Response to Planting–Fertilization Regimes

Except for the interaction between nitrogen application rate and nitrogen type in 2023 and the interaction between planting pattern and nitrogen type in 2024, which showed no significant effects on alfalfa yield (*p* > 0.05), all other main effects and interactions significantly influenced alfalfa yield (*p* < 0.05; Table 1).

Under the C application, alfalfa yield initially increased and then decreased with an increase in the nitrogen rate. Yields under BM and PM were significantly higher than under FP (Figure 1a,c). Within the same planting pattern, the yield of C2 in 2023 was significantly higher than those of N0, C1, and C3 by 54.21%, 24.20%, and 11.77%, respectively; in 2024, the corresponding increases were 50.83%, 21.67%, and 12.71%. At the same nitrogen level, BM and PM treatments in 2023 significantly increased alfalfa yield compared to FP by 43.56% and 30.07%, respectively; in 2024, the increases were 31.49% and 17.05%, respectively. Among all C treatments, BMC2 produced the highest alfalfa yield (two-year average: 24.84 t·hm^−2^), which was not significantly different from BMC3 and PMC2 but was significantly higher than FPN0, BMN0, and FPC2 by 97.11%, 59.63%, and 35.74%, respectively. Under the U application, alfalfa yields ranged from 13.34 to 21.06 t·hm^−2^ in 2023 and from 15.62 to 24.14 t·hm^−2^ in 2024. The yield patterns with an increase in the nitrogen rate and under different planting modes were generally consistent with those observed under C. Among all U treatments, BMU2 had the highest yield (two-year average: 22.60 t·hm^−2^), not significantly different from BMC2 and PMC2 but significantly higher than FPN0, BMN0, and FPU2 by 77.18%, 43.49%, and 30.33%, respectively. At the same nitrogen rate and planting mode, C application increased alfalfa yield by 8.46% compared with U (Figure 1b,d). Thus, the BMC2 treatment achieved the optimal yield performance.

### 2.2. Effects of Planting–Fertilization Regimes on Soil N_2_O Emissions

#### 2.2.1. N_2_O Emission Flux and Cumulative Emissions

Soil N_2_O emission flux exhibited a trend of initially increasing and then decreasing as the alfalfa growth stages progressed. In 2023, N_2_O flux ranged from 0.0042 to 0.1179 mg·m^−2^·h^−1^, with distinct emission peaks observed on 10 May, 26 June, and 10 August. In 2024, flux ranged from 0.0257 to 0.1401 mg·m^−2^·h^−1^, with peaks on May 15 and August 1 (Figure 2). Under the C application, the average N_2_O emission fluxes in PM and BM were 25.13% and 9.11% higher than in FP, respectively. Within the same planting pattern, C1, C2, and C3 treatments showed average increases of 4.82%, 12.23%, and 21.49% in the N_2_O flux compared with N0, respectively. Under the U application, N_2_O flux significantly increased after each top-dressing event. On average, the N_2_O flux under U was 24.61% higher than that under C. In PM and BM treatments, the average N_2_O fluxes were 24.08% and 10.01% higher than the values in FP, respectively. Within the same planting pattern, N_2_O fluxes in U1, U2, and U3 increased by 21.93%, 34.96%, and 63.31% compared to N0, respectively. Among all treatments, FPN0 had the lowest emission flux (two-year average: 0.04582 mg·m^−2^·h^−1^), while PMU3 had the highest (two-year average: 0.0897 mg·m^−2^·h^−1^).

Both the main effects of planting pattern, nitrogen fertilizer type, and nitrogen rate, as well as the interaction between nitrogen rate and fertilizer type, significantly affected cumulative N_2_O emissions (Table 2). Cumulative N_2_O emissions were lower under C than under U, increased with higher nitrogen rates, and were higher in BM and PM than in FP (Figure 3). Under C, cumulative N_2_O emissions in C1, C2, and C3 were 5.56%, 11.41%, and 20.01% higher than those in N0, respectively. At the same nitrogen level, PM and BM showed 21.31% and 6.73% higher cumulative emissions than FP, respectively. Under the urea application, cumulative N_2_O emissions in U1, U2, and U3 increased by 21.78%, 34.24%, and 61.47% compared to N0, respectively. At the same nitrogen level, cumulative emissions in PM and BM were 23.12% and 9.45% higher than those in FP, respectively. Among all treatments, PMC3, BMU3, and BMC2 had cumulative emissions 25.18%, 9.22%, and 38.83% lower, respectively, than the highest-emitting treatmentPMU3 (two-year average: 2.82 kg·hm^−2^).

#### 2.2.2. Soil Global Warming Potential and N_2_O Emission Intensity

The variation in soil global warming potential (GWP) under different nitrogen fertilizer types, application rates, and planting patterns was generally consistent with that of cumulative N_2_O emissions (Figure 4). Under the C application, GWP in C1, C2, and C3 was 5.86%, 12.26%, and 21.32% higher than that in N0, respectively. At the same nitrogen level, GWP in PM and BM was 23.27% and 7.97% higher than that in FP, respectively. Under the urea application, GWP in U1, U2, and U3 increased by 21.93%, 34.43%, and 61.91% compared with N0, respectively. At the same nitrogen level, GWP in FP was 18.95% and 9.18% lower than that in PM and BM, respectively.

Soil N_2_O emission intensity (GHGI) was lower under C than under U and followed a trend of initially decreasing and then increasing with the increase in the nitrogen rate. GHGI in BM was lower than in PM and FP (Figure 5). Under the C application, GHGI in BM was reduced by 20.15% and 19.49% compared to PM and FP, respectively. Within the same planting pattern, GHGI in C2 was reduced by 32.98%, 13.48%, and 17.07% compared to N0, C1, and C3, respectively. Under the U application, GHGI in BM was 19.07% and 16.79% lower than in PM and FP, respectively. Within the same planting pattern, GHGI in U2 was reduced by 3.52%, 9.89%, and 22.18% compared to N0, U1, and U3, respectively. Among all treatments, BMC2 had the lowest GHGI, with a two-year average of 0.0204 kg CO_2_-eq·kg^−1^.

### 2.3. Effects of Planting–Fertilization Regimes on Soil Environment

#### 2.3.1. Effects of Planting–Fertilization Regimes on Soil Temperature

Soil temperature in the 0~25 cm layer showed a decreasing trend with an increase in depth. The temperatures under PM (20.3~28.0 °C) and BM (19.1~26.7 °C) were higher than those under FP (18.7~25.1 °C) (Figure 6). At the same fertilizer type and nitrogen level, the average soil temperature at 5 cm depth under PM and BM was 11.37% and 4.89% higher than that under FP, respectively; similar increases were observed at 25 cm depth, where PM and BM were 11.37% and 4.89% higher than FP, respectively. No significant differences in soil temperature in the 0~25 cm layer were observed among different nitrogen fertilizer types or nitrogen application levels.

#### 2.3.2. Effects of Planting–Fertilization Regimes on Soil Moisture Content

Soil volumetric water content in the 0~30 cm layer ranged from 5.71% to 19.79%, with PM and BM treatments exhibiting higher values than FP. Soil moisture content increased with both the nitrogen application rate and soil depth (Figure 7). At the same fertilizer type and nitrogen rate, the average soil moisture content in PM and BM was 20.20% and 8.71% higher than that in FP, respectively. Under the FP planting pattern, the average soil moisture content in C1, C2, and C3 increased by 6.57%, 9.20%, and 20.03%, respectively, compared to N0; under the urea treatment, U1, U2, and U3 increased by 14.21%, 17.04%, and 26.89%, respectively, compared to N0. Under the BM pattern, the average soil moisture content in C1, C2, and C3 increased by 5.89%, 10.91%, and 15.75%, and in U1, U2, and U3 by 9.38%, 15.19%, and 16.94%, respectively, compared to N0. Under the PM pattern, the average soil moisture content in C1, C2, and C3 increased by 4.49%, 9.28%, and 4.94%, and in U1, U2, and U3 by 5.27%, 14.11%, and 17.80%, respectively, compared to N0. Among all treatments, PMU3 had the highest average soil moisture content (16.85%), which was 48.72%, 6.39%, and 12.17% higher than FPN0, FMU3, and PMN0, respectively.

#### 2.3.3. Effects of Planting–Fertilization Regimes on Soil Available Nitrogen Content

The effects of planting pattern, nitrogen fertilizer type, and nitrogen application rate as single factors, as well as the interactions between planting pattern and fertilizer type, and between fertilizer type and nitrogen rate, significantly influenced the soil nitrate nitrogen (NO_3_^−^-N) content. The effects of planting pattern and fertilizer type as main factors, as well as the interaction between planting pattern and nitrogen rate, significantly influenced the soil ammonium nitrogen (NH_4_^+^-N) content (Table 3).

Under the C application, both NO_3_^−^-N and NH_4_^+^-N contents increased with an increase in the nitrogen rate and were higher in PM and BM than in FP. Within the same planting pattern, compared to N0, the average NO_3_^−^-N contents in C1, C2, and C3 increased by 56.08%, 152.76%, and 160.95%, respectively, while NH_4_^+^-N contents increased by 73.59%, 118.25%, and 202.33%, respectively. At the same nitrogen rate, NO_3_^−^-N contents in PM and BM were 18.51% and 7.29% higher than in FP, and NH_4_^+^-N contents were 54.27% and 42.51% higher than in FP, respectively. Under the U application, soil NO_3_^−^-N and NH_4_^+^-N contents ranged from 10.33 to 20.78 mg·kg^−1^ and from 2.28 to 6.23 mg·kg^−1^, respectively, and their variation patterns with nitrogen rate and planting pattern were generally consistent with those observed under C. However, at the same nitrogen level and planting pattern, NO_3_^−^-N and NH_4_^+^-N contents under U were on average 18.32% and 17.04% higher, respectively, than those under C.

### 2.4. Correlation Between Soil N_2_O Emission Flux and Soil Environmental Factors

Correlation analysis showed that soil N_2_O emission flux was significantly positively correlated with soil moisture content, nitrate nitrogen (NO_3_^−^-N) content, and ammonium nitrogen (NH_4_^+^-N) content. Among these, the correlation between N_2_O flux and NO_3_^−^-N content was the strongest (*R*^2^ = 0.624), followed by the soil moisture content (*R*^2^ = 0.551) and NH_4_^+^-N content (*R*^2^ = 0.506). A non-significant negative correlation was observed between soil N_2_O emission flux and soil temperature (Figure 8).

### 2.5. Optimal Planting–Fertilization Regime for Yield Increase and Emission Reduction in Alfalfa

To evaluate the yield-increasing and emission-reducing effects of different planting–fertilization regimes in alfalfa grasslands, principal component analysis (PCA) was conducted based on eight indicators: soil temperature (T), soil moisture content (W), annual alfalfa yield (Y), cumulative soil N_2_O emissions (E), soil nitrate nitrogen content (N), ammonium nitrogen content (A), global warming potential (GWP), and greenhouse gas emission intensity (GHGI) (Table 4). The analysis showed that the eigenvalues of the first three extracted principal components were all greater than 1, with a cumulative contribution rate of 82.50%. Principal Component 1 accounted for 36.72% of the total variance (mainly reflecting the impact of GHGI); Principal Component 2 explained 29.28% of the variance (mainly associated with E, GWP, and Y); and Principal Component 3 explained 16.49% of the variance (mainly reflecting the influence of T).

Based on the comprehensive scores and ranking results (Figure 9), BM performed better than PM and FP; C was superior to U, with the best performance observed at the nitrogen application rate of 160 kg·hm^−2^. Among all treatments, BMC2 was identified as the optimal regime, while PMN0 was the least favorable.

## 3. Discussion

### 3.1. Effects of Planting–Fertilization Regimes on Alfalfa Yield

As a typical leguminous crop, alfalfa can fix atmospheric nitrogen via root nodule symbiosis. However, during early growth, post-cutting regrowth, and early spring under low temperatures, nitrogen fixation capacity is limited, making alfalfa dependent on soil nitrogen supply to meet its growth demand [31]. Exogenous nitrogen supplementation is an effective approach to enhance soil fertility and secure high alfalfa yield [32]. This study found that alfalfa yield increased and then decreased with an increase in the nitrogen application rate, peaking at 160 kg·hm^−2^. Similar trends were reported by Wang et al. [33] in wheat and Ma et al. [34] in cotton. Excessive nitrogen application can inhibit the activities of nitrate reductase and sucrose phosphate synthase in plants, disrupting nitrogen metabolism, weakening photosynthesis, and delaying reproductive growth [7]. However, Zhao et al. [35] reported that 120 kg·hm^−2^ was optimal for alfalfa in Henan, while Lu et al. [36] in Jiangsu found the highest yield at 225 kg·hm^−2^. These differences may be attributed to regional climatic conditions and soil nutrient status. Gansu, located in the arid Inland Northwest, is characterized by saline–alkali and infertile soils. Henan has moderate hydrothermal conditions and loamy soils with good fertility and nitrogen use efficiency. In Jiangsu, high rainfall causes nitrogen leaching losses up to 30%, and high temperatures accelerate nitrogen transformation, compounding losses. Nitrogen fertilizer type is another key factor affecting yield response. This study showed that a controlled-release nitrogen fertilizer application increased alfalfa yield by 8.46% compared to urea at the same nitrogen rate. Similar findings were reported by Belyaeva et al. [37] in ryegrass, Gao et al. [38] in maize, and Sun et al. [39] in rice. This is primarily due to the differences in nitrogen transformation rates [40]. Urea releases nitrogen rapidly, causing temporary nitrogen surges and losses via volatilization and leaching. In contrast, controlled-release nitrogen fertilizer uses coating technology to release nitrogen gradually, reducing NH_4_^+^-N and NO_3_^−^-N concentration spikes and stabilizing nitrogen supply [41]. However, Jin. et al. [42] found no significant yield difference between controlled-release nitrogen fertilizer and urea in winter wheat in the Huang–Huai–Hai region, possibly due to crop-specific growth traits. Alfalfa is harvested multiple times per year under high temperatures and a long growing season, likely making it more compatible with controlled-release nitrogen fertilizer release dynamics. In contrast, winter wheat has a dormant period and lower temperatures, limiting the effectiveness of controlled-release nitrogen fertilizer.

Planting pattern is also a critical agronomic factor affecting yield. Ridge mulching modifies soil moisture, nutrients, temperature, and aeration around the root zone, promoting plant growth and productivity [43]. This study found that BM and PM increased alfalfa yield by 23.49% and 18.65%, respectively, compared to FP. Similar yield enhancements due to mulching have been observed by Miceli et al. [44] in Italy, Samphire et al. [45] in the UK, and Lv et al. [46] in Gansu. DeVetter et al. [47] also found that biodegradable films outperformed polyethylene films in yield enhancement. Mulching improves soil solar radiation absorption, raises temperature, and redistributes moisture more effectively [48]. Compared to conventional plastic films, biodegradable films reduce late-season soil heat stress and prevent premature senescence [49]. As the film degrades, thermal insulation weakens, and lower nighttime temperatures reduce plant respiration losses, enhancing organic matter retention and yield [50]. Biodegradable mulch also retains moisture effectively in the early stages and develops micropores during degradation, reducing surface runoff and improving rainwater infiltration, which helps maintain optimal soil moisture during critical growth stages [51]. Furthermore, it avoids problems such as residual pollution and soil overheating associated with conventional plastic films, supporting sustainable yield improvement.

### 3.2. Effects of Planting–Fertilization Regimes on N_2_O Emissions in Alfalfa Grasslands

Nitrogen management affects N_2_O emissions by regulating microbial activity and nitrogen transformation processes [40]. This study showed that different nitrogen types and rates led to net N_2_O emissions in alfalfa fields. Under the urea application, significant N_2_O emission peaks appeared approximately 7 days after each top-dressing event. Similar patterns were reported by Oikawa et al. [52] in Sorghum bicolor. The hydrolysis of urea generates NH_4_^+^, which rapidly nitrifies into NO_3_^−^, causing spikes in inorganic nitrogen and triggering N_2_O release [53]. In contrast, controlled-release nitrogen fertilizer releases nitrogen more gradually, avoiding distinct emission peaks. In this study, controlled-release nitrogen fertilizer reduced N_2_O emissions by 17.95% compared to urea. However, Yao. et al. [54] found a 13.30% increase in N_2_O emissions under controlled-release nitrogen fertilizer in maize fields in the Songliao Plain. This discrepancy may be due to different application methods. In our study, controlled-release nitrogen fertilizer was applied once at green-up, allowing for a slow release throughout the growth period [55]. In contrast, Yao et al. [54] used an equal fertilization proportion with base fertilizer and top dressing to concentrate nitrogen supply in the middle of the corn growing season. However, this period coincides with the hot and rainy season, which may accelerate the release of nutrients from controlled-release nitrogen fertilizers. At the same time, the combination of fertilization disturbance and frequent rainfall can easily form anaerobic micro-zones, significantly enhancing denitrification and leading to increased N_2_O emissions [56]. Soil N_2_O emissions are closely related to environmental factors. This study found significant positive correlations between N_2_O flux and NO_3_^−^-N (*R*^2^ = 0.624) and NH_4_^+^-N (*R*^2^ = 0.506), consistent with results by Song et al. [57] in grasslands on the Qinghai–Tibet Plateau. The high availability of inorganic nitrogen promotes microbial nitrification and denitrification by providing substrates [58], and elevated NH_4_^+^-N and NO_3_^−^-N concentrations stimulate microbial abundance and activity, enhancing N_2_O emissions [51].

Planting patterns alter soil thermal and moisture regimes and nitrogen transformation processes, thus regulating the nitrification–denitrification intensity and N_2_O flux [59]. In this study, N_2_O emissions followed the order PM > BM > FP, with PM and BM increasing cumulative emissions by 23.12% and 9.45%, respectively. Similar findings were reported by Chen et al. [60] in garlic, Meng et al. [61] in potatoes in the Shiyang River Basin, and Samphire [45] in leeks in the UK. Plastic mulch retains more soil heat, enhancing organic nitrogen mineralization and microbial nitrification–denitrification [62,63]. Biodegradable films initially have similar insulation to plastic, but their degradation weakens this effect over time [64], thereby lowering N_2_O emission risk. In our study, N_2_O flux was negatively but non-significantly correlated with soil temperature (*R*^2^ = 0.095, *p* > 0.05). However, Chen et al. [65] found a positive correlation in tomato fields in Shaanxi, likely due to different crop types and climates. Soil temperatures during alfalfa growth (17.9~28.0 °C) were below the optimal range (25~35 °C) for nitrifiers [66], limiting microbial N_2_O production. Alfalfa’s deep roots may also improve soil aeration and enhance N_2_O reduction to N_2_ [67]. In contrast, tomato roots are shallower, and frequent irrigation in Shaanxi may promote denitrification [68]. Soil moisture is another important factor affecting N_2_O emissions under different planting regimes. This study found a significant positive correlation between soil moisture and N_2_O flux (*R*^2^ = 0.551, *p* < 0.05). However, Mumford et al. [69] found no such correlation in ryegrass fields in New South Wales. This may be due to water management differences. In our study, drip irrigation maintained a stable moist soil layer [70], which filled pore spaces, creating anaerobic conditions favorable for denitrification and N_2_O release [71,72]. In contrast, Mumford’s site experienced alternating wet–dry cycles, disrupting these processes and weakening the moisture–emission link [73].

GHGI is a key indicator for assessing the environmental efficiency of agricultural production, reflecting N_2_O emissions per unit of yield [74]. This study found that GHGI in BM was reduced by 19.07% and 16.79% compared to PM and FP, respectively. Similar results were reported by Chen et al. [75] in maize–wheat rotations and by Hu et al. [25] in rice in southern China. Although biodegradable mulch slightly increased GWP, the concurrent yield gain significantly reduced GHGI [40]. Both the nitrogen type and application rate affected GHGI. In the study by Zeng. et al. [76] on rice and Wang et al. [77] on lettuce, the authors found that C significantly reduced GHGI compared to urea. In this study, GHGI was lowest under C at 160 kg·hm^−2^. This is likely due to reduced leaching, smoother N cycling, and lower residual soil nitrogen. The BMC2 treatment had the lowest GHGI, indicating that combining C with biodegradable mulch effectively achieved yield gains and emission reductions. This is consistent with results by Chen et al. [26] in maize and Hu et al. [27] in rice.

Biological factors also play a key role in regulating yield and N_2_O emissions under different planting–fertilization regimes [78]. Future research should focus on the soil microbial community structure, functional microbial activity, and key enzymatic regulation of N_2_O production and transformation to clarify the mechanisms by which biological processes contribute to both yield and environmental benefits.

## 4. Materials and Methods

### 4.1. Experimental Site

The field experiment was conducted from April to September in both 2023 and 2024 at the Irrigation Experiment Station of the Gansu Jingtai Chuan Electric Power Lift Irrigation Water Resource Utilization Center (37°12′ N, 104°05′ E; elevation 1572 m). The site is located in a temperate arid continental climate zone, with an annual sunshine duration of 2652 h, an average annual solar radiation of 6.18 × 10^5^ J·cm^−2^, a frost-free period of 191 days, and a multi-year average evaporation of 2761 mm. During the experimental periods, the mean air temperature and total precipitation were 8.6 °C and 75.61 mm in 2023 and 12.28 °C and 279.87 mm in 2024, respectively. The test site is sandy loam soil. The surface soil (0~20 cm) had a bulk density of 1.61 g·cm^−3^, a pH of 8.11, and a field capacity of 24.10% (volumetric water content). The soil contained 6.09 g·kg^−1^ of organic matter, 1.62 g·kg^−1^ of total nitrogen, and 74.51 mg·kg^−1^ of available nitrogen.

### 4.2. Experimental Design

The tested alfalfa (*Medicago sativa* L.) cultivar was “Longdong alfalfa” (hereafter referred to as alfalfa), which was sown in April 2021. Before sowing, we inoculated lucerne seeds by soaking them in a solution containing highly effective rhizobacteria (*Sinorhizobia meliloti*). The land was leveled 10 days before sowing, and row sowing was used. The alfalfa sowing rate was 22.5 kg·hm^−2^. A completely randomized block design was used, with three factors: planting pattern, nitrogen fertilizer type, and nitrogen application rate. The three planting patterns (Figure 10) included flat planting without mulch (FP), ridge planting with biodegradable mulch (BM), and ridge planting with conventional plastic mulch (PM). The biodegradable mulch (BM, supplied by Shandong Tianzhuang Environmental Protection Technology Co., Ltd. (Jinan, China), with an induction period of 90 days) is light white and applied using a spray method. The plastic mulch (PM, supplied by Yangling Rui Feng Environ-mental Protection Technology Co., Ltd., (Yangling, China)) comprised polyethylene, with a thickness of 0.008 ± 0.001 mm, semi-transparent white in color, and a width of 100 cm. In the ridge cultivation plots, furrows were dug and ridges were formed, with a ridge height of 25 cm, a ridge width of 60 cm, and a ridge–furrow ratio of 1:1. Alfalfa was planted on the sides of the ridges and in the furrows, with four rows of alfalfa planted per ridge–furrow system, spaced 20 cm apart. In the flat-bed treatment plots, the alfalfa row spacing was 30 cm. Two nitrogen fertilizer types were applied: controlled-release nitrogen fertilizer (C, produced by Shandong Kingenta Ecological Engineering Group Co., Ltd. (Linyi, China); for polymer-coated controlled-release type, total nutrient content is 40%, N-P_2_O_5_-K_2_O mass fractions are 30%-4%-6%, respectively, coating thickness is 50 ± 5 μm, and particle size is 3–5mm.) and conventional urea (U; nitrogen content: 46.4%). Four nitrogen application rates were set: 0, 80, 160, and 240 kg·hm^−2^ (Table 5). A total of 21 treatment combinations were established, each with three replicates. The area of each plot was 42.9 m^2^ (5.5 m × 7.8 m). Fertilization and irrigation were performed via a drip irrigation system. In ridge-planted plots, the spacing between drip lines was 40 cm, whereas in flat-planted plots, it was 60 cm. Each dripper had a rated flow rate of 2 L·h^−1^. Water flow in each plot was regulated by valves and water meters (accuracy: 0.001 m^3^) installed on the irrigation pipes. Controlled-release nitrogen fertilizer was applied once as a basal dressing during the spring regrowth stage each year. Urea was applied in three split doses at a ratio of 6:2:2 before the first-cut regrowth stage (7 May in both 2023 and 2024), after the first harvest (20 June 2023, and 5 June 2024), and after the second harvest (6 August 2023, and 28 July 2024). Phosphorus fertilizer (single superphosphate, P_2_O_5_ content: 16%) and potassium fertilizer (potassium chloride, K_2_O content: 60%) were each applied at 50 kg·hm^−2^ as a one-time basal application before the first regrowth stage of each year. All cultivation and growth cycles involved full irrigation. The spring irrigation period for 2023 was April 20, and for 2024, it was April 25. Irrigation and other field management practices followed local standard alfalfa cultivation protocols. Alfalfa was harvested three times in 2023 (20 June, 29 July, and 12 September) and three times in 2024 (5 June, 28 July, and 25 September).

### 4.3. Measurement Items and Methods

#### 4.3.1. Yield

At the early flowering stage of each cutting, alfalfa was harvested to determine hay yield using a combination of fresh biomass measurement and the fresh-to-dry weight ratio method. A 1 m × 1 m quadrat was placed at the center along the diagonal of each plot. After harvesting, the fresh biomass within the quadrat was weighed (M_1_, kg). Next to the main quadrat, a smaller 50 cm × 50 cm sub-quadrat was randomly selected. After harvesting, the fresh biomass in the sub-quadrat was weighed (M_2_, g) and then sealed in a paper envelope, inactivated at 105 °C for 30 min, and oven-dried at 75 °C until constant weight. Once cooled, the dry weight was recorded (M_3_, g). The fresh-to-dry ratio was calculated as M_3_/M_2_. The total hay yield of all three cuttings was used to calculate the annual yield.

The alfalfa hay yield (Y, kg·hm^−2^) was calculated using the following equation:(1)$$Y=\frac{(M_{1}\times M_{2})\times 1000}{M_{2}}$$

#### 4.3.2. Greenhouse Gas N_2_O

Soil N_2_O was collected using the closed static chamber method [79]. The chamber measured 60 × 60 × 60 cm, with a base of 60 × 60 × 15 cm, and the underground part was inserted 10 cm into the alfalfa field. During sampling, the chamber was placed on the base and sealed with a water groove on the upper edge. The left side of the chamber was equipped with gas sampling and fan power interfaces. Two 12 V mini fans were installed at the upper left and lower right corners of the chamber to mix the gas, powered by external batteries. During sampling, the chamber was placed on the base; at other times, it was removed to ensure consistent light, air temperature, soil temperature, and humidity between inside and outside the chamber. Gas was sampled between 9:00 and 11:00 a.m. At 0, 10, 20, and 30 min, 40 mL of gas was collected using a 50 mL syringe, which was pumped in and out twice through the sampling port. Samples were immediately taken back to the laboratory and analyzed using a gas chromatograph (Shimadzu GC-2010 Pro, Kyoto, Japan). N_2_O was sampled every 15 days starting from the beginning of the experiment.

The N_2_O flux (F, mg·m^−2^·h^−1^) was calculated using the following equation [80]:(2)$$F=\rho \times H\times \frac{dc}{dt}\times \frac{273}{\left(273+T\right)}\times 60$$
where *ρ* is the density of N_2_O under standard conditions (*ρ* = 2 × 14/22.4 = 1.25, kg·m^−3^); *T* is the average temperature (°C) inside the closed chamber during sampling; *H* is the height of the chamber (m); *c* is the volume mixing ratio of N_2_O; and *dc*/*dt* is the rate of change in the N_2_O concentration during sampling (mg·m^−2^·min^−1^).

A positive value of F indicates that the soil is emitting N_2_O, while a negative value indicates N_2_O uptake by the soil.

The cumulative N_2_O emissions (M, kg·hm^−2^) were calculated using the following equation [81]:(3)$$M=\sum [(f_{i+1}+f_{i})\times 0.5]\times (d_{i+1}-d_{i})\times \frac{24}{100}$$
where *f* is the N_2_O flux (mg·m^−2^·h^−1^); *i* = 1, 2, … *n* is the sampling index; *d_i_*_+1_^−^ − *d_i_* is the number of days between two consecutive samplings; and 100 is the unit conversion coefficient.

The global warming potential (GWP, kg CO_2_-eq·hm^−2^) was calculated using the following equation [2]:(4)GWP=M×298
where *M* represents the cumulative N_2_O emissions during the measurement period (kg·hm^−2^).

The greenhouse gas emission intensity (GHGI, kg CO_2_-eq·kg^−1^) was calculated as(5)$$GHGI=\frac{GWP}{Y}$$
where *Y* is the yield (kg·hm^−2^).

#### 4.3.3. Soil Temperature

For PM and BM treatments, bent-stem thermometers were placed in the ridge–furrows of each plot. For the FP treatment, thermometers were placed between alfalfa rows in the center of each plot. During each N_2_O sampling event, soil temperature at a depth of 0~25 cm was measured at 5 cm intervals using bent-stem thermometers (produced by Wuqiang Hongxing Instrument Factory, Hebei Province, Hengshui, China).

#### 4.3.4. Soil Volumetric Water Content

For PM and BM treatments, one time-domain reflectometry (TDR) probe (150 cm in length) was installed in the ridge–furrow of each plot. For the FP treatment, one TDR probe was placed between alfalfa rows in the center of the plot. During each N_2_O sampling event, a portable soil profile moisture meter (PI-CO-BT, IMKO, Ettlingen, Germany) was used to measure soil volumetric water content at 0~30 cm depth at 10 cm intervals.

#### 4.3.5. Soil Nitrate and Ammonium Nitrogen

During each N_2_O sampling event, soil samples were collected using an “S”-shaped sampling method [82]. Soil NO_3_^−^-N and NH_4_^+^-N concentrations in the 0~10 cm soil layer were determined using 2 mol·L^−1^ KCl extraction (5 g air-dried soil; soil-to-solution mass ratio of 1:10), followed by analysis with a UV–visible spectrophotometer (T6 New Century, Beijing Puxi General Instrument Co., Ltd., Beijing, China).

### 4.4. Statistics and Analysis of Data

Microsoft Excel 2021 was used for data collation; Origin 2021 was used for graphing. IBM SPSS Statistics 27.0 was used for principal component analysis, and one-way ANOVA and Duncan’s multiple range test, as well as two-way ANOVA, were used to analyze the interaction effects. The significance level was set at *p* < 0.05.

## 5. Conclusions

(1) Compared with urea, controlled-release nitrogen fertilizer increased alfalfa yield by 8.46%. Ridge mulching with biodegradable film and plastic film increased alfalfa yield by 23.49% and 18.65%, respectively, compared with conventional flat planting. The BMC2 treatment achieved the highest yield, which was 97.11% and 35.74% higher than that under FPN0 and FPU2, respectively.

(2) Cumulative N_2_O emissions and greenhouse gas emission intensity (GHGI) under the C application were reduced by 23.89% and 25.84%, respectively, compared with urea. Both N_2_O emission flux and cumulative emissions increased with higher nitrogen application rates. PM and BM increased cumulative N_2_O emissions by 18.52% and 7.45%, respectively, compared to FP. The BMC2 treatment exhibited the lowest cumulative N_2_O emissions and GHGI.

(3) Soil N_2_O emission flux was significantly positively correlated with NO_3_^−^-N content, soil moisture content, and NH_4_^+^-N content.

(4) Principal component analysis showed that the BMC2 treatment significantly outperformed other treatments in improving yield and reducing N_2_O emissions and is therefore recommended as an optimal planting–fertilization regime for enhancing productivity and mitigating emissions in the Yellow River irrigation district of Gansu and similar ecological regions.

## Figures and Tables

**Figure 1 plants-14-02022-f001:**
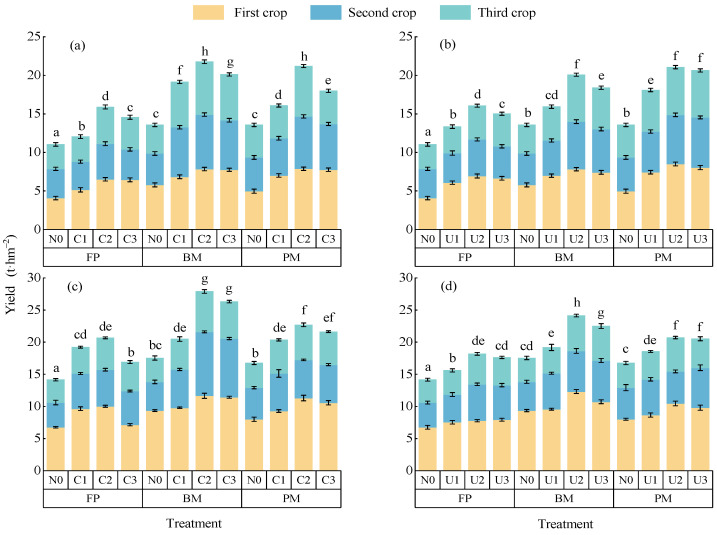
Impact of nitrogen application patterns on alfalfa yields. Note: Error bars indicate 95% confidence intervals. Different lowercase letters indicate significant differences between treatments (*p* < 0.05): (**a**,**c**) alfalfa yields with controlled-release nitrogen fertilizer applied in 2023 and 2024, respectively; (**b**,**d**) alfalfa yields with urea applied in 2023 and 2024. N0 represents no nitrogen application; C1, C2, and C3 represent controlled-release nitrogen fertilizer applications of 80 kg·hm^−2^, 160 kg·hm^−2^, and 240 kg·hm^−2^, respectively; and U1, U2, and U3 represent urea applications of 80 kg·hm^−2^, 160 kg·hm^−2^, and 240 kg·hm^−2^, respectively.

**Figure 2 plants-14-02022-f002:**
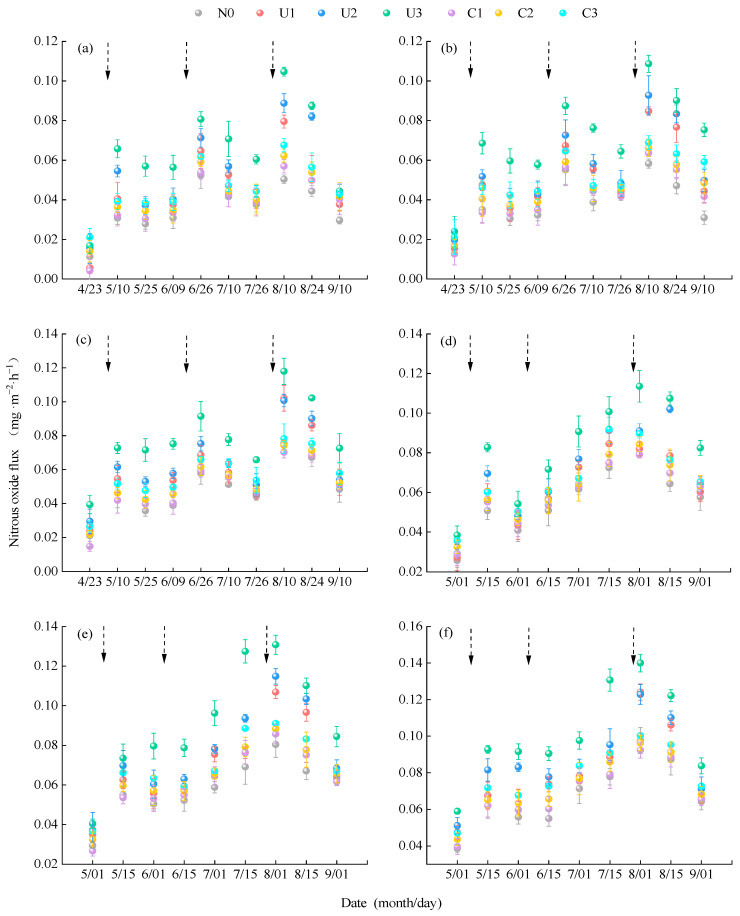
The impact of planting nitrogen application patterns on soil N_2_O emission fluxes. Note: Error bars and dots in the figure represent 95% confidence intervals and emission fluxes, respectively, and dashed arrows represent urea application: (**a**–**c**) flat planting without mulching, biodegradable film mulching, and plastic film mulching, respectively, in 2023; (**d**–**f**) flat planting without mulching, biodegradable film mulching, and plastic film mulching, respectively, in 2024. N0 represents no nitrogen application; C1, C2, and C3 represent controlled-release nitrogen fertilizer applications of 80 kg·hm^−2^, 160 kg·hm^−2^, and 240 kg·hm^−2^, respectively; and U1, U2, and U3 represent urea applications of 80 kg·hm^−2^, 160 kg·hm^−2^, and 240 kg·hm^−2^, respectively.

**Figure 3 plants-14-02022-f003:**
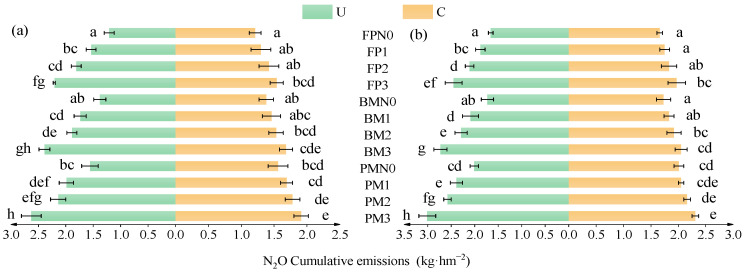
The impact of planting nitrogen application patterns on cumulative soil N_2_O emissions: (**a**,**b**) cumulative soil N_2_O emissions in 2023 and 2024, respectively. Error bars indicate 95% confidence intervals. Different lowercase letters indicate significant differences between treatments (*p* < 0.05).

**Figure 4 plants-14-02022-f004:**
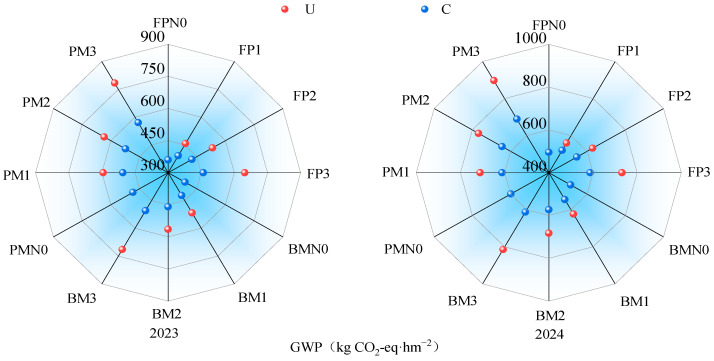
The impact of cropping nitrogen application patterns on GWP: GWP in 2023 and 2024, respectively.

**Figure 5 plants-14-02022-f005:**
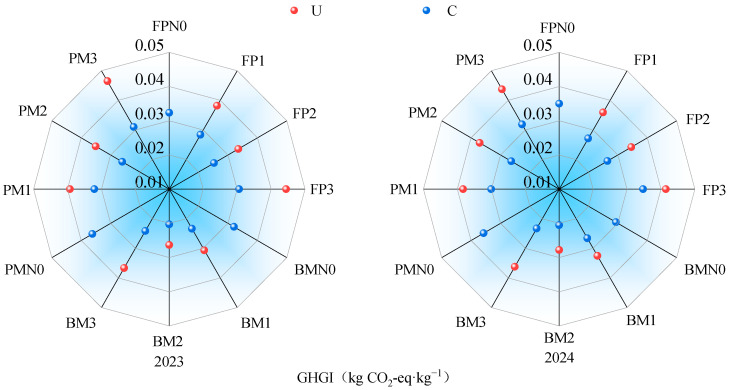
The impact of planting nitrogen application pattern on N_2_O emission intensity: GHGI in 2023 and 2024, respectively.

**Figure 6 plants-14-02022-f006:**
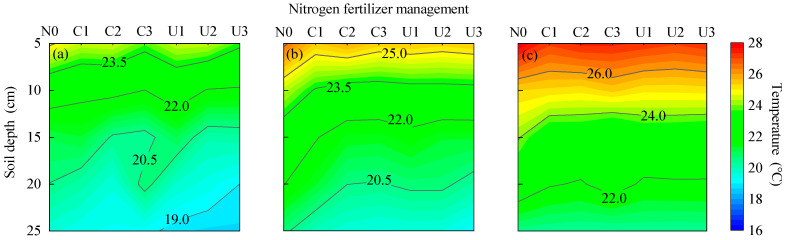
The impact of planting nitrogen application pattern on soil temperature: (**a**–**c**) flat planting without mulching, biodegradable film mulching, and plastic film mulching treatments, respectively.

**Figure 7 plants-14-02022-f007:**
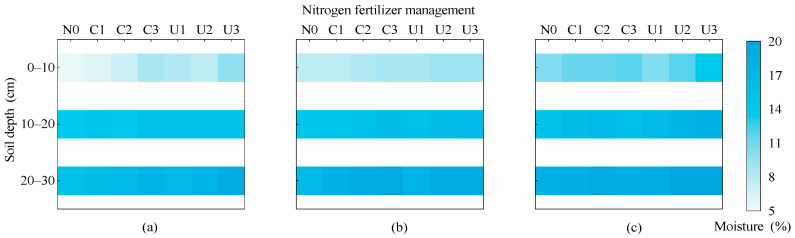
The impact of planting nitrogen application pattern on soil water content: (**a**–**c**) flat planting without mulching, biodegradable film mulching, and plastic film mulching treatments, respectively.

**Figure 8 plants-14-02022-f008:**
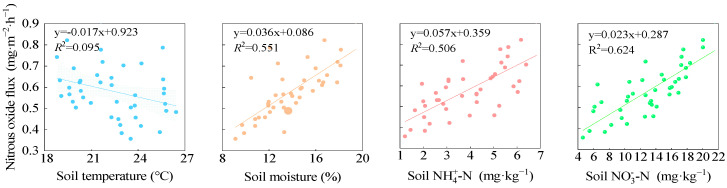
The correlation of soil N_2_O emission fluxes with soil environmental factors. Data in the figure are averaged; dots indicate N_2_O emission fluxes; straight lines indicate linear fit curves for N_2_O emission fluxes; shaded bands indicate 95% confidence bands for N_2_O emission fluxes.

**Figure 9 plants-14-02022-f009:**
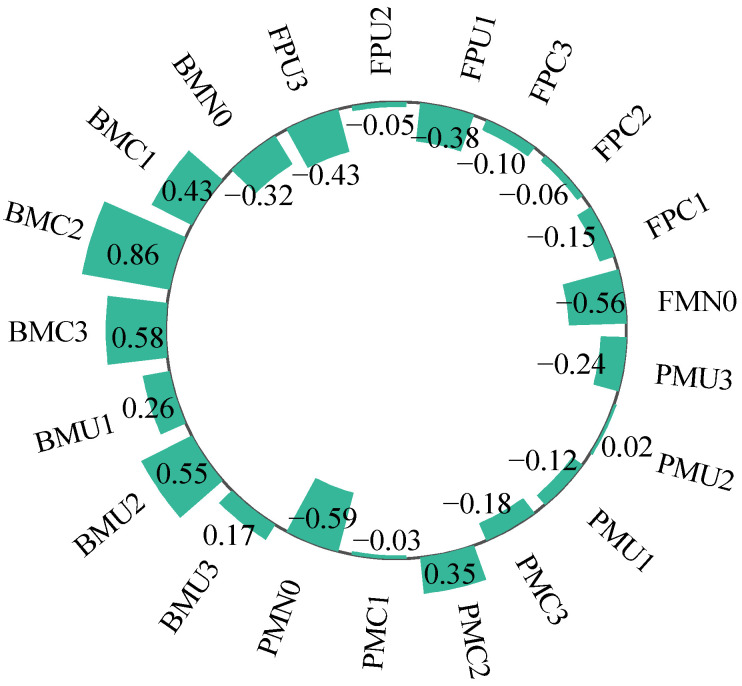
The composite score of yield increase and emission reduction of alfalfa grassland under the planting nitrogen application mode.

**Figure 10 plants-14-02022-f010:**
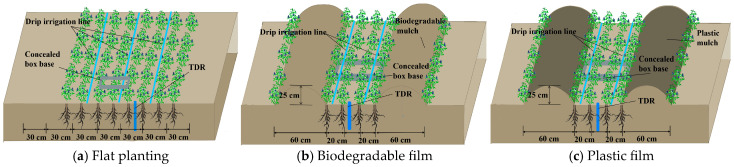
Schematic layout of the test plot.

**Table 1 plants-14-02022-t001:** Analysis of variance for the impact of planting nitrogen application patterns on alfalfa yield.

Year	M	N	B	M × N	M × B	N × B	M × N × B
2023	**	**	**	**	*	ns	**
2024	**	**	**	**	ns	**	*

In the table, M is the cropping pattern, N is the nitrogen application level, and B is the type of nitrogen fertilizer. **, *, and ns denote *p* < 0.01, *p* < 0.05 and *p* > 0.05, respectively.

**Table 2 plants-14-02022-t002:** Analysis of variance of soil N_2_O cumulative emissions by planting nitrogen application pattern.

Year	M	N	B	M × N	M × B	N × B	M × N × B
2023	**	**	**	ns	ns	*	ns
2024	**	**	**	ns	ns	**	ns

In the table, M is the cropping pattern, N is the nitrogen application level, and B is the type of nitrogen fertilizer. **, *, and ns denote *p* < 0.01, *p* < 0.05 and *p* > 0.05, respectively.

**Table 3 plants-14-02022-t003:** The effect of planting nitrogen application regime on the soil’s quick-acting nitrogen content.

Nitrogen Management	NO_3_^−^-N Content (mg·kg^−1^)			NH_4_^+^-N Content (mg·kg^−1^)		
	FP	BM	PM	FP	BM	PM
N0	5.38 ± 0.33 a	5.81 ± 1.11 a	6.25 ± 0.46 a	1.64 ± 0.19 a	1.72 ± 016 a	1.79 ± 0.25 a
C1	8.86 ± 0.46 b	9.35 ± 0.49 b	10.01 ± 0.41 b	2.36 ± 0.63 ab	2.96 ± 0.39 ab	3.02 ± 0.42 ab
C2	13.47 ± 0.60 d	15.25 ± 0.75 c	16.98 ± 1.52 c	3.69 ± 1.66 ab	4.21 ± 0.29 bc	4.67 ± 0.41 c
C3	14.84 ± 0.66 e	15.28 ± 0.71 c	17.06 ± 0.59 c	5.44 ± 0.23 bc	5.67 ± 0.41 ab	6.21 ± 0.69 d
U1	10.33 ± 0.58 c	12.86 ± 0.61 c	14.23 ± 0.58 c	2.28 ± 0.98 c	2.96 ± 0.18 d	3.32 ± 0.36 b
U2	14.21 ± 0.45 de	16.79 ± 1.38 c	17.22 ± 0.28 c	3.14 ± 0.68 bc	4.88 ± 0.63 cd	5.08 ± 0.71 cd
U3	17.56 ± 0.75 f	19.19 ± 0.24 d	20.78 ± 1.47 d	5.28 ± 0.77 c	5.75 ± 1.34 d	6.23 ± 0.92 d
M	**			**		
N	**			ns		
B	**			**		
M × N	*			*		
M × B	ns			ns		
N × B	**			ns		
M × N × B	ns			ns		

In the table, M is the cropping pattern, N is the nitrogen application level, and B is the type of nitrogen fertilizer. Different lowercase letters indicate significant differences between treatments (*p*< 0.05). **, *, and ns denote *p* < 0.01, *p* < 0.05 and *p* > 0.05, respectively.

**Table 4 plants-14-02022-t004:** Principal component factor loadings and variance contributions.

Ingredient	Factor Load		
	Principal Component 1	Principal Component 2	Principal Component 3
T	0.486	0.060	0.720
W	0.453	0.415	−0.324
Y	0.262	0.756	0.498
N	0.621	0.512	−0.426
A	0.585	0.516	−0.378
E	0.674	−0.729	−0.100
GWP	0.674	−0.729	−0.100
GHGI	0.890	−0.059	0.324
Eigenvalue	2.938	2.342	1.320
Variance contribution rate/%	36.723	29.28	16.498
Cumulative contribution rate/%	36.723	66.003	82.502

**Table 5 plants-14-02022-t005:** Field trial treatment design.

Cropping Patterns	Nitrogen Fertilizer Management		
	Nitrogen Fertilizer Types	Nitrogen Application Levels (kg·hm^−2^)	Application Rate Between First, Second and Third Crops
Flat planting without mulching (FP)	Urea (U)	80	6:2:2
		160	6:2:2
		240	6:2:2
	Controlled-release nitrogen fertilizer (C)	80	1:0:0
		160	1:0:0
		240	1:0:0
	—	0	—
Biodegradable film (BM)	Urea (U)	80	6:2:2
		160	6:2:2
		240	6:2:2
	Controlled-release nitrogen fertilizer (C)	80	1:0:0
		160	1:0:0
		240	1:0:0
	—	0	—
Plastic film (PM)	Urea (U)	80	6:2:2
		160	6:2:2
		240	6:2:2
	Controlled-release nitrogen fertilizer (C)	80	1:0:0
		160	1:0:0
		240	1:0:0
	—	0	—

## Data Availability

All data supporting this study are included in the article.

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
