# Peer review of "Yield Increase and Emission Reduction Effects of Alfalfa in the Yellow River Irrigation District of Gansu Province: The Coupling Mechanism of Biodegradable Mulch and Controlled-Release Nitrogen Fertilizer"

_plants, 2025, doi:10.3390/plants14132022_

Round 1
Reviewer 1 Report
Comments and Suggestions for Authors
See attachment

Author Response
We are grateful to your valuable comments and suggestions. These comments are very valuable and helpful for revising and improving our manuscript, and provide important guidance for our research. We carefully considered every point raised and revised the manuscript with reference to each valuable comment. Your comments are laid out below in italicized font and specific concerns have been numbered. Our responses are given in normal font, the modified content are given in bold font, and changes to the manuscript are given in revision mode. We provide a detailed explanation as follows:
Comments 1: Fig. 9 is not necessary. Just add one data in the text. The alfalfa was inoculated with Rhizobium? If not indicate if there are previous knowledge about the occurrence of enough Rhizobium in the soil. The soil has very low N concentration. this fact could be indicated as a reason for N fertilization.
Response 1: Thank you for your careful review of our manuscript and your valuable suggestions for improvement. We agree with your opinion and have deleted Fig 9 from the revised manuscript. The relevant description of meteorological data on page 14, section4.1, lines 445-447, states: “During the experimental periods, the mean air temperature and total precipitation were 8.6 °C and 75.61 mm in 2023, and 12.28 °C and 279.87 mm in 2024, respectively”. This adjustment makes the manuscript more concise while retaining key information.
Regarding the suggestion on rhizobia, we have added a more detailed description on L 456 of Section 4.2 on P14 in the manuscript. The added sentence is:“Before sowing we inoculated lucerne seeds by soaking them in a solution containing highly effective rhizobacteria (Sinorhizobia meliloti)”.
Thank you again for your valuable feedback.
We would like to express our sincere thanks to the reviewer for constructive feedback. We have carefully addressed each point and believe that the revisions have significantly improved the manuscript.
Looking forward to your early reply.
Best regards,
Yours sincerely,
Wenjing Chang.

Reviewer 2 Report
Comments and Suggestions for Authors
This manuscript reports on a well designed and conducted two year field experiment to evaluate the effect of planting patterns, mulching, N source, and N rates on the productivity of alfalfa and N greenhouse gas emissions. The manuscript is well organized and written, and the results are well presented, analyzed and discussed.
I suggest that additional details be provided in the Materials and Methods section to improve the reproducibility of the study (see suggestions below). In the introduction may want to include a brief definition of ridge planting with a description of how it is incorporated into crop production systems. Additional specifications of the particular ridge planting system used in the current study may be included in the M&Ms sections (see comments below for L 435).
In the Materials and Methods section it may also be helpful to specify the final planting density (No plants/Hectare) for each of the treatments (ridge vs flat systems as described in Fig. 10), to clarify whether planting density variations may have represented a confounding effect in the study.
Were alfalfa tissue N levels assessed as part of the study? The evaluation of tissue N levels would have provided an additional reference point, to assess the effect of treatments, N form, and rates on N dynamics and gas emissions.
Additional comments on the text include,
L 142 & 176, Figures 1 & 2, Do you need to indicate in the legend what the treatment acronyms/abbreviations stand for (C, U, N0)?
L 274, Should this be labeled as Figure 8 (Figure 7 has already been presented, L 232)?
L 363-367, Please rephrase, unclear,
L 435-436, Please provide a more detailed description of the ridge planting system. Does this consist of raised beds or ridges (as compared to flat planting). If so, describe the height and width of the ridges, and distance from center to center. Describe if alfalfa is planted on top or bottom of the ridges.
L 435-437, Is it possible to provide a description (specifications, such as material used, color, and thickness) of both the biodegradable and plastic mulches? This is important to improve the reproducibility of the study. The color of the mulch will also have an effect on the soil temperature, below the mulch.
L 440, Is it possible to provide specifications of the slow release fertilizer used (composition e.g. coating material, as discussed in L 324-325)?
L 476, Should you include a reference here, re: ‘static chamber method’?
Minor edit suggestions are included in the attached copy of the manuscript.
////

Author Response
We are grateful to your valuable comments and suggestions. These comments are very valuable and helpful for revising and improving our manuscript and provide important guidance for our subsequent research. We carefully considered every point raised and revised the manuscript with reference to each valuable comment. The reviewer comments are laid out below in italicized font and specific concerns have been numbered. Our responses are given in normal font, the modified contents are given in bold font, and changes to the manuscript are given in revision mode. We provide a detailed explanation as follows:
Comments 1: I suggest that additional details be provided in the Materials and Methods section to improve the reproducibility of the study (see suggestions below). In the introduction may want to include a brief definition of ridge planting with a description of how it is incorporated into crop production systems. Additional specifications of the particular ridge planting system used in the current study may be included in the M&Ms sections (see comments below for L 435).
Response 1: Thank you for your valuable suggestions regarding the introduction section. We have added explanations regarding the definitions and application methods of ridge cultivation and mulching techniques in lines 87-97 of the original manuscript. The specific additions are as follows: “Designing structure of ridge-furrow is a key aspect of ridge cultivation technology. Zhang et al. [19] found that when the ridge-furrow ratio was 60 cm: 60 cm, the rainwater collection efficiency was significantly higher than that of traditional earth ridges on the Loess Plateau, China. Liu et al. [20] found that when the ridge-furrow ratio was 75 cm: 50 cm, it could increase the photosynthetic rate and yield of wheat on the Huang Huaihai Plain, China. Ridge cultivation combined with plastic mulching integrates ridge cultivation and plastic mulching, effectively reducing soil moisture evaporation, promoting the convergence of rainfall on the ridge surface into the furrows, increasing the infiltration depth and water-holding capacity of soil moisture in the furrows, and forming a synergistic mechanism of dynamic water exchange and complementarity between ridges and furrows.”
Two additional references have been added:
[19] Zhang, X.; Wang, R.; Liu, B.; Wang, Y.; Yang, L.; Zhao, J.; Xu, J.; Li, Z.; Zhang, X.; Han, Q. Optimization of ridge–furrow mulching ratio enhances precipitation collection before silking to improve maize yield in a semi–arid region. Agric. Water Manage. 2023, 275, 108041.
[20] Liu, K.; Shi, Y.; Yu, Z.; Zhang, Z.; Zhang, Y. Improving photosynthesis and grain yield in wheat through ridge–furrow ratio optimization. Agronomy 2023, 13, 2413
In our study, the ratio of ridges to furrows is 1:1. The specific specifications have been added to lines 470–474 of Section 4.2 on P14. The revised content has been highlighted in red for your review.
Comments 2: In the Materials and Methods section, it may also be helpful to specify the final planting density (No plants/Hectare) for each of the treatments (ridge vs flat systems as described in Fig. 10), to clarify whether planting density variations may have represented a confounding effect in the study.
Response 2: Thank you for your rigorous review of the methodological details in our manuscript. Your suggestion regarding whether the planting density (especially the difference between ridge and flat systems) affects the research results is very important. We have made the following additions to lines 458-459 of the “Materials and Methods” section: “The alfalfa grassland for the experiment was established in April 2021. The land was leveled 10 days before sowing, and strip sowing was used. The alfalfa sowing rate was 22.5 kg·hm⁻2 for all treatments.”
Comments 3: Were alfalfa tissue N levels assessed as part of the study? The evaluation of tissue N levels would have provided an additional reference point, to assess the effect of treatments, N form, and rates on N dynamics and gas emissions.
Response 3: Thank you for your valuable suggestions on nitrogen cycle analysis. Regarding the assessment of nitrogen content in alfalfa tissue, we would like to clarify that since this study primarily focuses on nitrogen transformation processes at the soil-gas interface (such as the correlation between N₂O emission fluxes and soil available nitrogen content), the experimental design did not systematically measure nitrogen content in alfalfa plants. We fully agree with your perspective that plant nitrogen uptake data can enhance the explanatory power of nitrogen fate, and it is an important direction for future research improvements. The current data has been validated through the effects of treatment on soil inorganic nitrogen dynamics (NH₄⁺ and NO₃⁻ content), with detailed results presented in Section 2.3.3.
Comments 4: L 142 & 176, Figures 1 & 2, Do you need to indicate in the legend what the treatment acronyms/abbreviations stand for (C, U, N0)?
Response 4: Thank you for your valuable feedback. Regarding the issue you raised about the clarity of the abbreviations (C, U, N0) in Figures 1 & 2, we have added their full definitions to the figure captions based on your comments. “N0 represents no nitrogen application, C1, C2, and C3 represent controlled-release nitrogen fertilizer applications of 80 kg·hm⁻2, 160 kg·hm⁻2, and 240 kg·hm⁻2, respectively, and U1, U2, and U3 represent urea applications of 80 kg·hm⁻2, 160 kg·hm⁻2, and 240 kg·hm⁻2, respectively.”
Comments 5: L 274, Should this be labeled as Figure 8 (Figure 7 has already been presented, L 232)?
Response 5: Thank you for your careful review of the details of the paper. Your point regarding the chart numbering on line 292 is entirely correct; it should be Figure 8. We have corrected the incorrect numbering and updated all subsequent chart numbers and references in the manuscript accordingly. The revised version is highlighted in revision mode.
Comments 6: L 363-367, Please rephrase, unclear.
Response 6: Thank you for your careful review of our manuscript and your valuable suggestions for improvement. We apologize for the unclear wording in lines 363-367 that you pointed out and have reworked the relevant content. The revised wording for “In contrast, Yao’s study applied controlled-release nitrogen fertilizer in two equal splits (basal + topdressing) during midseason high temperatures and rainfall, accelerating nutrient release and promoting anaerobic microsites via fertilization disturbance and frequent precipitation, thereby enhancing denitrification and N₂O emissions [54]” is as follows: “Yao et al. [54] used an equally fertilization proportion with base fertilizer and top dressing to concentrate nitrogen supply in the middle of the corn growing season. However, this period coincides with the hot and rainy season, which may accelerate the release of nutrients from controlled-release nitrogen fertilizers. At the same time, the combination of fertilization disturbance and frequent rainfall can easily form anaerobic micro-zones, significantly enhancing denitrification and leading to increased N₂O emissions.”
Comments 7: L 435-436, Please provide a more detailed description of the ridge planting system. Does this consist of raised beds or ridges (as compared to flat planting). If so, describe the height and width of the ridges, and distance from center to center. Describe if alfalfa is planted on top or bottom of the ridges.
Response 7: Thank you very much for your professional comments on our manuscript. Your feedback has been extremely helpful to us. We have added specific parameters regarding the ridge cultivation system in Section 4.2 of the manuscript. The revised sections are marked in red, and the modifications are as follows: “In the ridge cultivation plots, furrows were dug and ridges were formed, with a ridge height of 25 cm, a ridge width of 60 cm, and a ridge-to-furrow ratio of 1:1. Alfalfa was planted on the sides of the ridges and in the furrows, with four rows of alfalfa planted per ridge-furrow system, spaced 20 cm apart; In the flat-bed treatment plots, the alfalfa row spacing was 30 cm.”
Comments 8: L 435-437, Is it possible to provide a description (specifications, such as material used, color, and thickness) of both the biodegradable and plastic mulches? This is important to improve the reproducibility of the study. The color of the mulch will also have an effect on the soil temperature, below the mulch.
Response 8: Thank you for your valuable feedback. We have added detailed specifications for the mulch film in the Materials and Methods section: “The biodegradable mulch (BM, supplied by Shandong Tian Zhuang Environmental Protection Technology Co., Ltd., with an induction period of 90 days) is light white in color and applied using a spray method; the plastic mulch (PM, supplied by Yangling Rui Feng Environ-mental Protection Technology Co., Ltd.) made of polyethylene, with a thick-ness of 0.008 ± 0.001 mm, semi-transparent white, and a width of 100 cm.”
Comments 9: L 440, Is it possible to provide specifications of the slow-release fertilizer used (composition e.g. coating material, as discussed in L 324-325)?
Response 9: Thank you for your detailed review of the experimental details. We have provided complete specification details regarding controlled-release nitrogen fertilizer. We have updated lines 479-481 on P15 of the “Materials and Methods” section: “C, produced by Shandong Kingenta Ecological Engineering Group Co., Ltd.; for polymer-coated controlled-release type, total nutrient content is 40%, N-P₂O₅-K₂O mass fractions are 30%-4%-6%, respectively, coating thickness is 50±5μm, and particle size is 3-5mm.”
Comments 10: L 476, Should you include a reference here, re: ‘static chamber method’?
Response 10: Thank you for your concern regarding the methodological rigor of our manuscript. Regarding the description of the static box method, we have added the following a key reference. We have inserted the citations in line 518 which are highlighted in red in the manuscript:
[80] Zhao, Z.J.; Hao, Q.J.; Tu, T.T.; Hu, M.L.; Zhang, X.Y.; Jiang, C.S. Effect of ferric-carbon micro-electrolysis on greenhouse gas emissions from constructed wetlands. Environ. Sci. 2021, 42, 3482-3493.
Comments 11: Minor edit suggestions are included in the attached copy of the manuscript.
Response 11: Thank you for your careful review of the details of our manuscript. We have carefully reviewed all your editorial suggestions in the attached copy of the manuscript and revised grammar, spelling, references, etc. The specific revisions are as follows:
(1) We have changed “King of Forages” in line 112 of the introduction to “Queen of Forages”.
(2) The name of Figure 1 has been changed from “Impact of planting nitrogen application patterns on alfalfa yields” to “Impact of nitrogen application patterns on alfalfa yields”.
(3) The misspelling of “under” in line 220 has been corrected to “Under”.
(4) All incorrectly labeled figure numbers have been checked and corrected.
(5) The P12 document “Alfalfa is harvested multiple times per year under high temperatures and a long growing season, which making it more compatible with controlled-release nitrogen fertilizer release dynamics” has been changed to “Alfalfa is harvested multiple times per year under high temperatures and a long growing season, likely making it more compatible with controlled-release nitrogen fertilizer release dynamics”.
(6) The P12 document “Biodegradable mulch also retains moisture effectively in early stages and develops micropores during degradation” has been changed to “Biodegradable mulch also retains moisture effectively in the early stages and develops micropores during degradation”.
(7) The P12 document “Similar patterns were reported by Oikawa et al. [52] in seabuckthorn” has been changed to “Similar patterns were reported by Oikawa et al [52] in Sorghum bicolor”.
(8) Change “Medicago sativa L.” in line 455 to “Medicago sativa L.”.
(9) Change “Drip irrigation belt” in the figure to “Drip irrigation line”. Also, indicate the ridge height.
(a) Flat planting |
(b) Biodegradable film |
(c) Plastic film |
(10) Change the abbreviation “CRF” for controlled-release nitrogen fertilizer in line 578 of the conclusion to “C”.
(11) Change “A field study on CO2, CH4 and N2O emissions from rice paddy ang impact factors” to “A field study on CO2, CH4 and N2O emissions from rice paddy and impact factors” in the references.
Thank you again for your valuable feedback.
We would like to express our sincere thanks to the reviewer for constructive feedback. We have carefully addressed each point and believe that the revisions have significantly improved the manuscript.
Looking forward to your early reply.
Best regards,
Yours sincerely,
Wenjing Chang.

Reviewer 3 Report
Comments and Suggestions for Authors
Very well written overall - just a few comments:
p13 what is soil type of study area?
p14 how much water was applied per unit land area? when did irrigation start in the spring? was it the same for all growth cycles?
p16 "During each N₂O sampling event, soil samples were collected using an "S"-shaped sampling method." What is 'S' shaped sampling method (give citation)?
Author Response
We are grateful to your valuable comments and suggestions. These comments are very valuable and helpful for revising and improving our manuscript and provide important guidance for our subsequent research. We carefully considered every point raised and revised the manuscript with reference to each valuable comment. The reviewer comments are laid out below in italicized font and specific concerns have been numbered. Our responses are given in normal font, the modified contents are given in bold font, and changes to the manuscript are given in revision mode. We provide a detailed explanation as follows:
Comments 1: p13 what is soil type of study area?
Response 1: Thank you for your careful review of the research background. Regarding the soil information for the study area on P13, line 448, we have provided a complete explanation: “the test site is sandy loam soil”. The surface soil (0~20 cm) had a bulk density of 1.61 g·cm⁻2, a pH of 8.11, and a field capacity of 24.10% (volumetric water content). The soil contained 6.09 g·kg⁻1 of organic matter, 1.62 g·kg⁻1 of total nitrogen, and 74.51 mg·kg⁻2 of available nitrogen.
Comments 2: p 14 how much water was applied per unit land area? when did irrigation start in the spring? was it the same for all growth cycles?
Response 2: Thanks for your positive comments. We have added a note on irrigation quotas in the Materials and Methods section to ensure a more rigorous experimental design. “All cultivation and growth cycles are fully irrigated. The spring irrigation period for 2023 is April 20, and for 2024, it is April 25. Irrigation and other field management practices followed local standard alfalfa cultivation protocols.” The revisions are on P14, lines 495-497 of the updated manuscript, which we have highlighted in red.
Comments 3: p 16 "During each N₂O sampling event, soil samples were collected using an "S"-shaped sampling method." What is 'S' shaped sampling method (give citation)?
Response 3: Thank you for your interest in sampling methods. The “S-shaped sampling method” is a standardized soil spatial sampling strategy, with the following specific operational guidelines: Within each plot, 5-8 sampling points are uniformly distributed along an “S”-shaped path (avoiding field edges and areas with obvious anomalies). Soil samples from each point are mixed to form a representative plot sample. This method effectively reduces the impact of soil spatial heterogeneity. We have inserted the relevant reference on P17, line 561.
[83] Li, M.W.; Zhu, Q.H.; Liu, H.; Xia, X.M.; Huang, D.Y. Method for detecting soil total nitrogen contents based on pyrolysis and artificial olfaction. Int J Agric & Biol Eng. 2022, 15, 167-176.
Thank you again for your valuable feedback.
We would like to express our sincere thanks to the reviewer for constructive feedback. We have carefully addressed each point and believe that the revisions have significantly improved the manuscript.
Looking forward to your early reply.
Best regards,
Yours sincerely,
Wenjing Chang.
